# The implications of parent mental health and wellbeing for parent-child attachment: A systematic review

Alixandra Risi[1]*, Judy A. Pickard[1], Amy L. Bird[1,2]

1 School of Psychology and Early Start, University of Wollongong, Wollongong, New South Wales, Australia,
2 School of Psychology, University of Waikato, Hamilton, New Zealand

☯ These authors contributed equally to this work.
* anr652@uowmail.edu.au

**Data Availability Statement:** All relevant data are within the manuscript and its Supporting information files.

**Funding:** The authors received no specific funding for this work.

## Abstract

### Background

Parent mental health and wellbeing may have implications for understanding attachment transmission. In this systematic review, we synthesise the published literature to determine the nature of the relationship between parent mental health and wellbeing and the intergenerational transmission of attachment and to provide recommendations for future research, clinical practice and intervention.

### Method

Using the preferred reporting items for systematic reviews and meta-analyses (PRISMA) approach, five electronic databases were searched for peer-reviewed empirical studies, published in English. Articles were considered for inclusion if data was collected on adult attachment, child attachment, and a domain of parent mental health/wellbeing. No date parameters were applied to the search strategy. The review was registered with PROPSERO (registration number: CRD42020157247).

### Results

Eleven studies examining the impact on parent mental health and wellbeing on the intergenerational transmission of attachment were identified for inclusion in this review. Our review found preliminary evidence that parent mental health and wellbeing play a role in the intergenerational transmission of attachment. Other key findings from the review were: evidence quality is mixed due to variable measurement of attachment and mental health; studies have mostly included correlational analysis or do not utilise contemporary methodological approaches to testing mediating or moderating relationships; and literature is largely focused on psychopathology and negative factors of mental health.

### Conclusions

The limited scope of parent mental health and wellbeing constructs examined in the literature, the sparse use of robust statistical analyses, and the lack of literature in general makes

**Competing interests:** The authors have declared that no competing interests exist.

it difficult to draw conclusions on how and why parent mental health impacts attachment transmission. Addressing these limitations will further progress attachment-related literature and may have particular implications for attachment-informed interventions with clinical populations.

## Introduction

Attachment is a construct of "lasting psychological connectedness" that is transmitted from one generation to the next [1, p.195]. A notable proportion of variance in the transmission of attachment security, however, remains unexplained [2]. We know that mental health is related to attachment security [3], but less is known about how mental health, and more contemporary conceptualizations of wellbeing, might help to understand attachment transmission. There is currently a lack of clarity in the literature regarding: how mental health is conceptualized; the ways parent mental health and adult and child attachment are related; and the ways attachment is measured. To make progress towards better understanding attachment transmission and informing future literature and intervention, a more systematic approach to understanding the relationship between parent mental health and adult and child attachment is needed. This systematic review will outline a number of different models to demonstrate the ways mental health and wellbeing may be related to adult and child attachment. It will also adopt a wider conceptualization of mental health beyond specific diagnoses and examine the impact of positive wellbeing on the intergenerational transmission of attachment.

Underpinned by cognitive and dynamic models of relationships, attachment theory [4] proposed that an individual's early experiences of care form internal working models, which in turn provide relatively stable conceptualizations of the world, self and others that inform later behavior. Bowlby [4] described attachment as an integrative process, through which infants internalise the experiences of caregiving to develop their own internal capacities for self-soothing and emotional regulation. An individual is likely to develop a secure attachment style when their early experiences provide safety and security through the availability and accessibility of a sensitive caregiver [5, 6]. Contrarily, the absence of positive and sensitive early caregiving experiences increases the propensity for developing an insecure attachment style. Studies consistently confirm a link between caregiver sensitivity and attachment security [3, 7–9]. Three insecure attachment styles have been identified: insecure-avoidant, insecure-ambivalent, or insecure-disorganised [5, 10]. Individual differences in attachment during infancy are observed through Ainsworth's [6] seminal paradigm, the Strange Situation Paradigm (SSP). The SSP assesses infant responses to being placed in a novel environment where they are introduced to a stranger and separated from their primary caregiver [6]. Initially proposed by Bowlby [1] and confirmed by subsequent studies, the attachment bonds infants create with their primary caregivers during infancy and early childhood play a vital role in their psychological, emotional, social, and cognitive development. The continuity of attachment style is then supported by the strategies developed for regulating and relating with others that are carried forward into new contexts, forming the foundation of an individual's life-long attachment style [5, 11, 12].

A core hypothesis of attachment theory is that patterns of attachment are transmitted from one generation to the next [13]. The development of the Adult Attachment Interview [AAI; 14] provided, for the first time, a valid and reliable window into the parent's attachment state of mind. Using the AAI, Main et al. [15] demonstrated a link between a mother's attachment

representations as coded from responses to the AAI and her infant's attachment to her. The AAI is a semi-structured interview exploring childhood experiences with attachment figures and the meaning the person currently gives to those experiences [14]. Rather than relying on potentially biased self-reports of relationship patterns, the AAI was designed to 'surprise the unconscious' and is regarded as the gold standard measure of adult attachment [16]. Adult classifications of attachment derived from the AAI (i.e., secure, preoccupied, dismissive and unresolved/cannot classify) parallel those identified in childhood and support the continuity of attachment history [17]. Further to early findings related to the antecedents of individual differences in infant attachment by Ainsworth et al. [5], Main et al. [15] posited that a mother's state of mind with respect to attachment guides her sensitive behavior toward her infant which in turn influences infant attachment quality. The intergenerational transmission of attachment has been observed in diverse populations including community samples [18], low-socioeconomic-status populations [19], adolescent mothers [20], mothers with clinical depression [21], and across cultures [22–25]. However, at the end of a decade of research examining attachment transmission, van IJzendoorn's [2] seminal meta-analysis found that parental sensitivity explained less than 25% of the variance linking maternal representations and infant attachment [see also 26, 27]. van IJzendoorn [2] termed the unexplained variance the 'transmission gap', referring to the aspect or aspects of the attachment transmission process from parent to infant which cannot be fully explained by parental sensitivity. Despite significant research efforts over the past 25 years, limited progress in explaining the transmission gap has been made [13].

However a recent meta-analysis synthesising three decades of research provides a more nuanced and perhaps complicated view on the intergenerational transmission of attachment [28, 29]. Attachment transmission appears weaker in samples with known risks (e.g., adolescent mothers, parents with psychopathology, families from low-socioeconomic backgrounds) and stronger when attachment was measured beyond infancy [28, 29]. Additionally, the way attachment is measured poses another difficulty for attachment research. While the SSP and AAI are the gold standard measures for assessing infant and adult attachment respectively [30], naturalistic observation and self-report measures have also been developed such as the Attachment Q Sort [ASQ; 31] and the Attachment Style Classification Questionnaire [32]. However, different attachment measures emphasize different attachment phenomena [33]. Evidence suggests that not all attachment measures actually assess the same construct, despite often being termed 'attachment measures' [33, 34]. After finding attachment transmission was weaker in younger than older children, Verhage et al. [28] hypothesised that the way attachment was measured may have confounded the moderating effect of child age on attachment transmission. Estimating the true size of the transmission gap and understanding factors other than sensitivity associated with attachment transmission, may vary according to the attachment measure utilised.

The transmission gap has therefore been described as "one of the most perplexing issues" facing attachment researchers during the past two decades [35, p.3]. Attempts to understand it have focused largely on the measurement of maternal behavior with some focus on more complex interactions between cognitive, physiological, social, ecological, and genetic constructs. There is also compelling evidence that genetics are unlikely to account for the transmission gap. Twin studies have consistently shown negligble genetic contribution to attachment security [e.g., 33, 34, 36, 37]. Dozier et al. [38] found no difference in the rate of intergenerational attachment transmission among foster case parent-infant dyads and biological parent-infant dyads. Taken together, these studies seem to indicate genetic mechanisms may not be an answer to closing the transmission gap.

Attachment researchers have suggested the examination of parent mental health may provide insight into the intergenerational transmission of attachment [e.g., 6, 35, 39]. For example,

attachment insecurity has repeatedly been found to be associated with the unresponsive, rejecting, and insensitive parenting that occurs at higher rates in parents experiencing depression [40]; leading to the suggestion that maternal depression may have an adverse impact on mother-infant attachment. Similarly, past experience of trauma may interfere with a parent's ability to respond sensitively and effectively to their infant's needs, thus compromising the development of secure parent-child attachment [41].

Traditionally, conceptualizations of mental health focused primarily on illness and disorder [42]. Contemporary conceptualizations, however, extend beyond the deficit-focused and ill definitions of the past and recognise mental health lies along a continuum with positive wellbeing [42, 43]. Redefining mental health highlights the importance of studying both psychopathology and positive wellbeing trans-diagnostic factors of mental health when considering the mechanisms that underlie parenting and relational development [44]. Indeed, research has suggested positive wellbeing constructs that should be considered, including self-compassion, mindfulness, and mentalization [e.g., 39, 41, 42].

Attachment and mindfulness are strongly correlated with one another and with similar wellbeing outcomes [47]. Mindfulness can be defined as the non-judgemental self-regulation of attention that involves experiencing the full range of emotional, cognitive and physiological experiences without avoidance [48]. From an attachment perspective, facilitating mindful awareness may allow a caregiver to be more open and responsive to present moment interactions with their infant [49]. This may support caregivers with their own history of insecure attachment to essentially 'override' unhelpful internal representations and experience new ways of relating [50]. Self-compassion refers to a state of mind that involves both higher levels of self-kindness, mindfulness, and common humanity and lower levels of self-judgment, isolation, and over-identification [51]. Parent self-compassion is associated with the activation of a caregiving mentality [52] and maternal warmth [53]. Thus, self-compassionate caregivers may be better able to engage in sensitive and responsive parenting, facilitating the development of a secure attachment. Mentalization–or reflective functioning as operationalisation of mentalization–refers to the ability to understand and interpret the behavior in terms of underlying mental states in oneself and in others [54]. Although not conceptualized as a mental health construct per se, mentalization is critical to psychological wellbeing [55]. The importance of mentalization in different areas of psychopathology has been demonstrated in research, with studies finding lower mentalizing capacities being related to psychiatric illness compared to normal controls [54, 56]. Parental mentalizing, the propensity of caregivers to "show frequent, coherent, or appropriate appreciation of their infants' internal states" [55, p.1245], is suggested to promote a secure attachment relationship as the parent is able to make sense of their child's mental state and respond appropriately [50]. Meta-analytic research has shown mentalization contributes to explaining the variance in infant attachment security [57], with a number of researchers [e.g., 13, 56] suggesting this construct may be promising to bridge the transmission gap.

To provide a comprehensive picture of attachment transmission, it is important to go beyond establishing whether a relationship exists with parent mental health and wellbeing to understanding the precise mechanisms of how and why. If indeed parent mental health and wellbeing impacts the intergenerational transmission of attachment, there are several ways in which this might occur.

In mediation, a third variable explains the relationship between the independent and dependent variables [58]. Mediation has historically been tested empirically using Baron and Kenny's [59] 'causal steps' regression analysis principles. More recent analytic approaches such as structural equation modelling and bootstrapping however, allow mediating mechanisms to be tested without causal steps. Such analyses are statistically more rigorous because they tend to have the highest power and the best Type I error control [60], and are more robust to smaller sample

sizes [61]; leading to methodologists generally no longer recommending Baron and Kenny's approach [58]. When testing mediation, longitudinal research designs enable mediation effects to be tested in a more rigorous manner than cross-sectional designs [62].

The mediation model is the premise for the intergenerational transmission of attachment and, in turn, the transmission gap: sensitive caregiving and other potential variable/s are the mechanisms through which adult attachment influences child attachment. Research using this model to examine the effect of maternal depression on attachment transmission in a community sample of mothers and their adult daughters found maternal depression mediated the influence of mothers' attachment insecurity on daughters' attachment insecurity [40]. Constructs in line with the contemporary conceptualization of mental health also appear to have been investigated using mediation. For example, a mediational analysis revealed that when maternal reflective functioning was controlled for, the relationship between adult and infant attachment disappeared [46], suggesting that maternal reflective functioning may be a central mechanism in the intergenerational transmission of attachment.

If mental health or wellbeing does indeed act as a mediator or mechanism in the relationship between parent attachment state of mind and infant attachment, it is unlikely to do so on its own. Instead, mental health or wellbeing may be part of a pathway that is related to differences in caregiving behavior. From a statistical perspective, this is described as serial mediation (Fig 1), whereby there are two or more mediators with one influencing the other. For example, we know that an insecure or unresolved adult attachment state of mind is associated with increased rates of psychopathology [63] (see path *a* in Fig 1). Parental psychopathology in turn is associated with poorer quality caregiving [40], which is associated with higher rates of insecure or disorganized infant attachment [15] (see path *b* in Fig 1). Indeed, Iyengar et al. [41] has suggested parental experience of trauma may function in this way, as trauma may interfere with a parent's ability to sensitively respond to their infant resulting in insecure parent-child attachments.

Moderation is another model which may also help to clarify associations of parent and child attachment and parent mental health (Fig 2). A moderator is a variable that affects the strength or direction of the relationship between the independent and dependent variable [58]. Applied to attachment transmission, the strength of the relationship between adult attachment and child attachment (i.e., the rate of transmission of attachment) may differ depending on parent mental health.

In research using the moderation model, Tarabulsy et al. [20] found maternal depression did not act as a moderator of the association between maternal state of mind about attachment and infant attachment security in a sample of adolescent mother-infant dyads. The findings of Tarabulsy et al. [20] and Besser and Priel [40], which differed when depression was examined as a moderating variable and mediating variable respectively, demonstrate the importance of distinguishing between how mental health constructs are associated with attachment

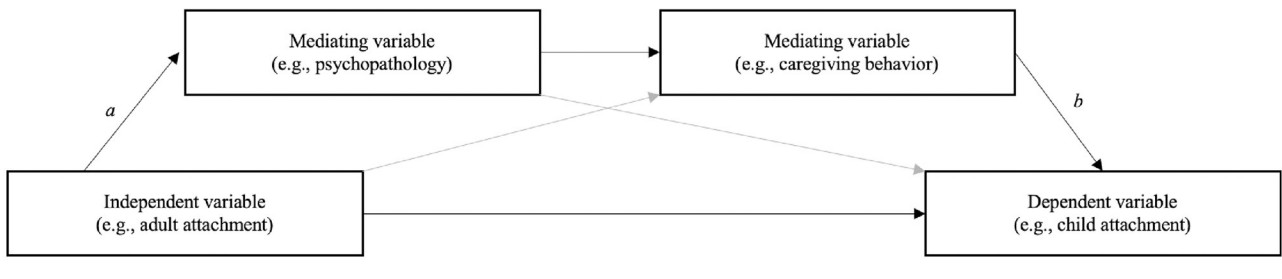

**Fig 1. Serial mediation model.**

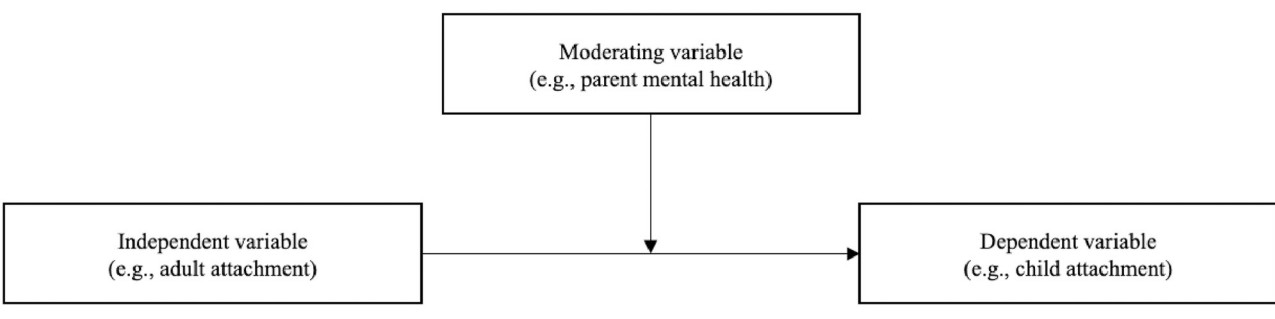

**Fig 2. Moderation model.**

transmission so a more comprehensive understanding can be achieved. Alternatively, moderation by attachment could also be considered, whereby parent attachment state of mind may moderate the relationship between parent mental health and infant attachment. Using this model, McMahon et al. [21] found the relationship between maternal depression and child attachment was moderated by maternal attachment state of mind in a sample of middle-class mother-infant dyads.

A final way to look at the relationship between the intergenerational transmission of attachment and parent mental health is to solely examine correlations among a specific high-risk population (e.g., parents that have experienced trauma) rather than directly testing mediation or moderation. In correlation, an association exists between variables and predictions can be made about one from the other [64]. While the correlational model gives us information about what happens to attachment transmission within a high-risk group, it does not directly test the moderation by mental health model that a more varied sample could. Using the correlational model with a sample of mothers who had experienced childhood abuse and neglect and their infants, Berthelot et al. [45] found over three-quarters of infants had an insecure attachment, indicative of intergenerational transmission of attachment.

The examination of mental health and wellbeing constructs as mechanisms for attachment transmission is important not only to extend the attachment transmission literature but also to inform future research and clinical intervention. Through the current systematic review, we aim to provide a comprehensive overview of the current literature examining how and in what way parent mental health might impact the intergenerational transmission of attachment. Through the process, we also aim to answer a number of specific questions about the literature:

1. What constructs of parental mental health are examined in the attachment transmission literature, and how are they measured?

2. How has attachment been measured within this literature?

3. What statistical models have been used within this literature?

4. Can the relationship between adult attachment and child attachment be explained by mediation and/or moderation models using parent mental health?

## Methods

### Protocol

The protocol for the current study was registered with the International Prospective Register of Systematic Reviews (PROSPERO, registration number: CRD42020157247). The search

strategy used to identify articles for inclusion in the review was in accordance with the Preferred Reporting Items for Systematic Review and Meta-Analysis (PRISMA) guidelines for reviews [65] (Fig 3). No sources of funding were provided for this systematic review.

## Search strategy

Studies included in this review were identified by searching online databases and reference lists of identified articles between July and August 2020. An online database search was made of five databases including PsychINFO, Web of Science, MEDLINE, Scopus, and SocINDEX. The search strategy incorporated two concepts: the transmission of adult attachment to parent-child attachment and parental experience of trauma and mental health. Search terms were (attach* transmission or transmission of attach* or parent-child attach* or parent-infant attach* or mother-infant attach*) AND (trauma or PTSD or adverse childhood experiences or complex PTSD or mental health or mental illness or depression or anxiety or stress or personality disorder* or mindfulness or mentaliz* or reflective function* or self-compassion).

No date parameters were placed on the search strategy. All records were imported into End-Note (Version X9) and then Covidence, a systematic review management website. Articles considered for inclusion were limited to non-duplicated articles published in English in peer-reviewed journals. Inclusion and exclusion criteria were applied to remaining articles. Titles and abstracts were screened to identify studies with a focus on the impact of parent mental health on the transmission of adult attachment to parent-child attachment. Review papers and

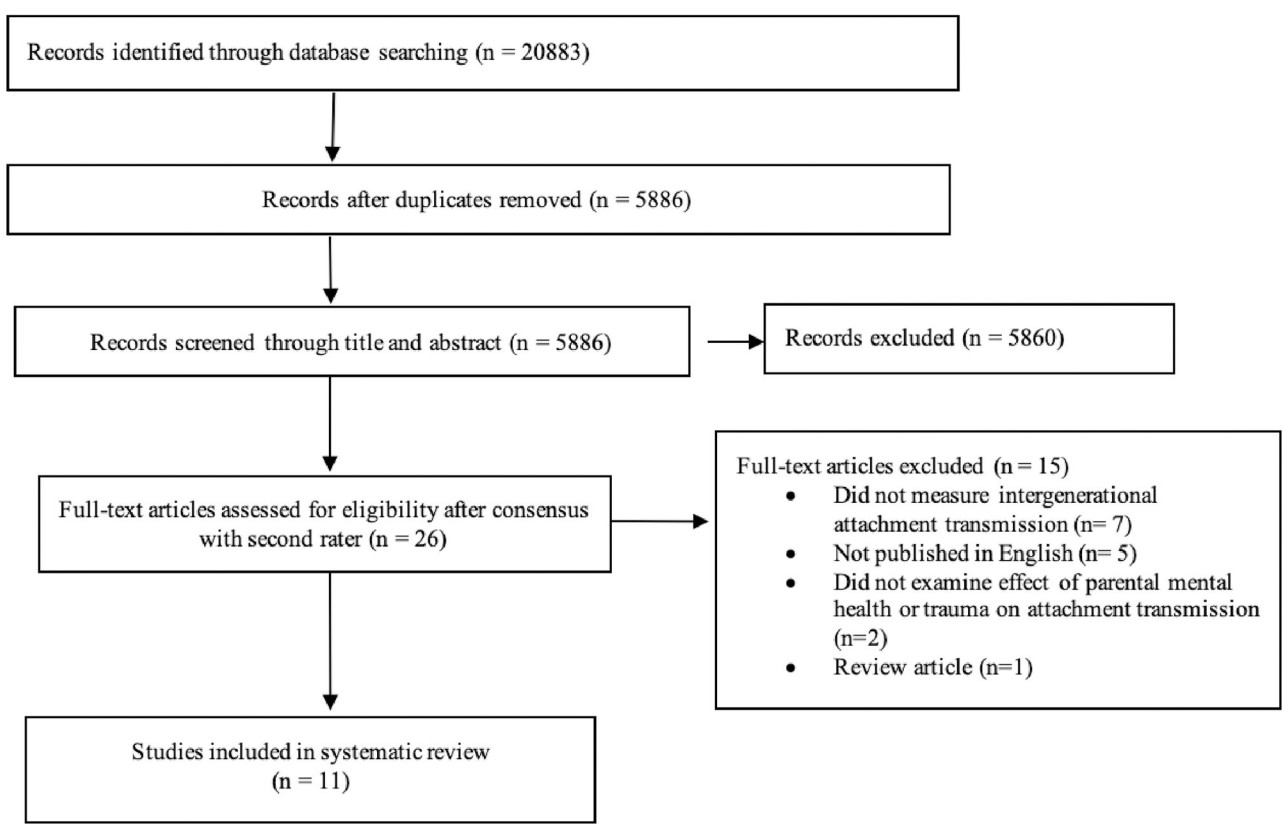

**Fig 3. PRISMA flowchart for study identification and selection process.**

studies examining the efficacy of an intervention were removed. The reference lists of articles being considered for review were searched to determine eligibility.

## Study selection

Articles were considered for inclusion in the current review provided that they met the following eligibility criteria:

- Published in English within a peer reviewed journal.

- Examined attachment transmission from parent to child.

- Considered/examined impact of parent mental health (e.g., anxiety, depression, trauma, personality disorder, reflective functioning, mentalization, mindfulness) on the transmission of attachment.

- Measures were included to assess adult attachment, child attachment, and at least one domain of mental health or trauma.

- The purpose of the study did not include evaluating of an intervention.

- The purpose of the study was not to review literature.

To provide a comprehensive overview of the current literature, all articles that reported they were assessing attachment, regardless of the actual measure used, were considered for inclusion in the review. The primary focus of this review was to identify the role of parent mental health and wellbeing on the transmission of adult attachment to parent-child attachment without the influence of and exposure to a treatment, program, or other type of intervention, hence intervention studies were excluded. This decision was made in consultation with other reviews within the parent mental health field [e.g., 63].

## Quality assessment and data extraction

Two members of the research team independently conducted a formal assessment of article quality using the *Appraisal of Cross-Sectional Studies* (AXIS). The AXIS is a quality appraisal tool that was specifically designed to critically assess the quality and risk of bias in cross-sectional studies for systematic reviews. The tool was developed by an expert panel using the Delphi methodology in consultation with current literature [66]. The tool comprises of 20 items relating to the identification of research aims, appropriateness of study design, use of valid measures and statistical analyses, and consideration of bias. Despite the measure originally being developed for cross-sectional studies, the items are relevant for both the cross-sectional and longitudinal studies included in the current review. The checklist design of the AXIS does not provide a cut-off numerical score for study eligibility. Unlike other quality assessment tools, the AXIS tool allows users the flexibility of a subjective assessment of overall quality, risk of bias, and quality of reporting. The only published study that investigated the psychometric properties of the AXIS demonstrated the tool has poor overall inter-rater reliability and moderate concurrent validity [67]. The authors highlight, however, that the generalisability of their findings is limited due to their use of 'advanced novice' raters and articles specific only to the topic of health-related quality of life and breast cancer. The AXIS has been effectively used by other reviews within the parental mental health field [68]. To our knowledge there is currently no gold standard tool for assessing the quality of observational studies [67]. Hence, despite its limitations, the AXIS was considered appropriate for the current review as it attempts to address the shortcomings in other available tools.

Following quality assessment, the first author extracted information from included studies relating to study aims, participant information, study design, assessment time points, location, measures, data analyses and key results. A second researcher within the research team supervised this process.

# Results

## Literature search

A total of 20,883 articles were identified through electronic database searching (n = 20,883), with 5886 studies remaining for screening after removing duplicates (n = 14,997). Articles were screened by title and abstract to identify empirical studies that examined the impact of parental mental health or trauma on the transmission of adult attachment to child-infant attachment. Based on their abstracts, a total of 26 articles appeared to meet the inclusion criteria and were included in the full-text review (n = 26). Two reviewers independently conducted full-text reviews. Following discussion between reviewers, 15 studies were excluded, in accordance with the eligibility criteria, leaving 11 studies for inclusion in the final review. Fig 3 illustrates this process.

The first and third author independently completed the AXIS for the remaining 11 studies to determine the quality of the articles. While no numerical cut-off value is required by the AXIS, caution should be taken when interpreting articles which met minimal criteria (Table 1). All studies met at least 7 of the 20 criteria. All studies identified clear aims, had an appropriate design, adequately described the basic data, and defined statistical significance. All but one study used appropriate outcome variables and valid measures. Karakas et al. [69] provided incorrect information on the procedure for coding the SSP, which limits our capacity in being able to judge whether this measure was used in a valid way. Additionally, they used the

**Table 1. AXIS quality assessment appraisal for studies included in the systematic review.**

| | | 1 | 2 | 3 | 4 | 5 | 6 | 7 | 8 | 9 | 10 | 11 |
|---|---|---|---|---|---|---|---|---|---|---|---|---|
| Introduction | Clear aims | ✓ | ✓ | ✓ | ✓ | ✓ | ✓ | ✓ | ✓ | ✓ | ✓ | ✓ |
| Methods | Appropriate design | ✓ | ✓ | ✓ | ✓ | ✓ | ✓ | ✓ | ✓ | ✓ | ✓ | ✓ |
| | Sample size justified | X | X | X | X | ✓ | X | X | X | X | X | X |
| | Population defined | ✓ | ✓ | ✓ | X | ✓ | X | ✓ | ✓ | ✓ | ✓ | ✓ |
| | Sample size representative of population | ✓ | ✓ | ✓ | X | ✓ | X | ✓ | ✓ | ✓ | ✓ | ✓ |
| | Selection process representative | ✓ | X | ✓ | X | X | X | X | ✓ | ✓ | X | ✓ |
| | Measures to address non-responders | ✓ | X | X | X | X | X | ✓ | ✓ | X | X | X |
| | Appropriate outcome variables | ✓ | ✓ | ✓ | ✓ | X | ✓ | ✓ | ✓ | ✓ | ✓ | ✓ |
| | Valid measures | ✓ | ✓ | ✓ | ✓ | X | ✓ | ✓ | ✓ | ✓ | ✓ | ✓ |
| | Defined statistical significance | ✓ | ✓ | ✓ | ✓ | ✓ | ✓ | ✓ | ✓ | ✓ | ✓ | ✓ |
| | Methods described | ✓ | ✓ | X | ✓ | X | X | X | ✓ | ✓ | ✓ | ✓ |
| Results | Result data described | ✓ | ✓ | ✓ | ✓ | ✓ | ✓ | ✓ | ✓ | ✓ | ✓ | ✓ |
| | Response rate does not raise concerns about non-response bias | ✓ | X | X | X | X | X | ✓ | ✓ | X | X | X |
| | Non-responder information described | ✓ | X | X | X | X | X | X | ✓ | X | X | X |
| | Results internally consistent | ✓ | ✓ | X | ✓ | ✓ | X | X | ✓ | ✓ | ✓ | X |
| | Results presented for analyses | ✓ | X | X | ✓ | ✓ | X | X | ✓ | ✓ | X | X |
| Discussion | Conclusions justified | ✓ | ✓ | X | ✓ | X | X | ✓ | ✓ | ✓ | ✓ | ✓ |
| | Limitations identified | ✓ | ✓ | ✓ | ✓ | ✓ | X | ✓ | ✓ | ✓ | ✓ | X |
| Other | No funding or conflicts of interest | ✓ | ✓ | ✓ | ✓ | ✓ | ✓ | ✓ | ✓ | ✓ | ✓ | ✓ |
| | Ethical approval/consent attains | X | X | ✓ | ✓ | ✓ | X | X | ✓ | ✓ | X | X |

Parental Bonding Instrument (PBI) to evaluate a mother's attachment with her own parent. The PBI is a measure of parenting behaviors rather than a measure of a parent's own attachment. Nine studies clearly used a sample that was representative of the reference population however it was difficult to ascertain whether the remaining studies used a representative sample. For example, Besser and Priel [40] reported participants responded to their call for volunteers but did not identify exactly where or what this call was. One study provided sample size justification while three studies undertook measures to address and describe non-responders. Five studies identified if they had ethical approval or attained the consent of participants.

## Overview of included studies

In total, 11 articles were included in the systematic review of the originally identified 5886 articles. All articles contained original quantitative data. A total of 649 mothers and 72 fathers were included, equalling 721 parents across the studies. Parent age ranged from 13.5–78 years. A total of 721 of the parents' children were included. A total of seven studies included infant children, while one study included children aged in their middle childhood to early adolescence. One study included children aged over 16 years, and two studies included adult children. The characteristics of these studies are shown in Table 2. Five of the studies employed a cross-sectional design, while the other six were longitudinal studies. Studies took place in the United States of America (n = 3), Turkey (n = 3), Canada (n = 2), Israel (n = 2), and Australia (n = 1). The date of publication of the articles ranged from 2003 to 2020. Sample sizes ranged from 44 parent-child dyads to 111 dyads ($M = 67.7$, $SD = 22.8$). There was no observed pattern in sample size based on location of publication. With the exception of three studies who utilised clinical populations, the majority of studies included participants from community samples. Of the eight articles with community samples, seven were high-risk community samples.

**Table 2. Overview of the included studies.**

|  |  | N | % |
|---|---|---|---|
| Study design | Cross-sectional | 5 | 45.5 |
|  | Longitudinal | 6 | 54.5 |
| Data collections points (for longitudinal studies) | Two | 3 | 50 |
|  | Three | 2 | 33.3 |
|  | Six | 1 | 16.6 |
| Population | Community | 8 | 72.7 |
|  | Clinical | 3 | 27.3 |
| Variables | Trauma | 3 | 27.3 |
|  | Depression | 3 | 27.3 |
|  | Psychiatric disorder | 1 | 9.1 |
|  | Reflective functioning | 1 | 9.1 |
|  | Substance abuse | 1 | 9.1 |
|  | Bipolar disorder | 1 | 9.1 |
|  | Trauma-specific reflective functioning | 1 | 9.1 |
| Measure | Self-report | 11 | 100 |
|  | Observational | 6 | 54.5 |
| Location | USA | 3 | 37.3 |
|  | Turkey | 3 | 27.3 |
|  | Canada | 2 | 18.2 |
|  | Israel | 2 | 18.2 |
|  | Australia | 1 | 9.1 |

Additional participant characteristics included adolescent mothers and parents from low socioeconomic backgrounds. Outcome variables included trauma (n = 3), depression (n = 3), psychiatric disorders (n = 1), substance use (n = 1), bipolar disorder (n = 1), parental reflective functioning (n = 1), and trauma-specific reflective functioning (n = 1). All studies employed self-report measures (n = 11), while six studies also included observation measures (interview and clinician rated measure). Across the 11 studies, five different measures were used to assess adult attachment and three different measures were used to assess child attachment. For overview, clarity and as in other attachment systematic reviews [e.g., 70], we divided attachment measurement methodology into two categories: direct measurement (e.g., observational measures) and indirect measurement (e.g., self-report measures). Six articles used direct measure of adult attachment, while five articles used direct measures of child attachment. Five articles used direct measures of attachment for both adult and child attachment. Indirect measures of adult attachment included the Relationship Questionnaire (n = 1), the Adult Attachment Style Classification Questionnaire (n = 2), the Parental Bonding Instrument (n = 1), and the Adult Attachment Scale (n = 1). Indirect measures of child attachment included the Children's Attachment Style Classification Questionnaire (n = 1) and the Attachment Behaviour Q-Sort (n = 1). Studies that examined adult children utilised measures of attachment for adults. A number of screening tools and assessment tools to measure outcome variables were used across the studies of which a summary is reported in Table 3. A summary of the characteristics and results of the studies included in the systematic review are presented in Table 4.

## Statistical analyses

The majority of studies used regression analyses (n = 7) and correlations (n = 6) for the purpose of statistical analyses. Other statistical analyses utilised included chi-square tests (n = 4), ANOVAs (n = 3), mediation (n = 2), Fisher's Exact Tests (n = 2), and t-tests (n = 1). Mediation analyses were tested using either Baron and Kenny's (1986) assumptions (n = 1) or structural equation modelling methods (n = 1).

**Table 3. Screening tools and measures of the included studies.**

| Variable | Measure | Acronym | Number |
|---|---|---|---|
| Adult Attachment | Adult Attachment Interview | AAI | 6 |
| | Relationship Questionnaire | RQ | 1 |
| | Adult Attachment Style Questionnaire | | 2 |
| | Parental Bonding Instrument | PBI | 1 |
| | Adult Attachment Scale | AAS | 1 |
| Child Attachment | Strange Situation Paradigm | SSP | 6 |
| | Children's Attachment Style Questionnaire | | 1 |
| | Attachment Q Sort | AQS | 1 |
| Trauma | Adult Attachment Interview (coded for trauma) | AAI | 2 |
| | Adverse Childhood Experience Questionnaire | ACEQ | 1 |
| Depression | Center for Epidemiological Studies Depression Scale | CES-D | 3 |
| | Composite International Diagnostic Interview Depression Module | | 1 |
| Psychiatric illness | Diagnostic and Statistical Manual of Mental Disorders, 4th Edition | DSM-4 | 1 |
| Bipolar disorder | Structured Clinical Interview for DSM | SCID | 1 |
| Parental reflective functioning | Parent Development Interview | PDI | 1 |
| Trauma-specific reflective functioning | Adult Attachment Interview (coded according to the Reflective-Functioning Manual for Application to Adult Attachment Interview) | AAI | 1 |

**Table 4. Characteristics and results of the studies included studies.**

| Author/s | Aims | Design | Location | Sample | N | Adult (parent) attachment measure | Child attachment measure | Trauma/ mental health measures | Statistical analysis performed | Key results |
|---|---|---|---|---|---|---|---|---|---|---|
| 1. Berthelot et al., 2015 [45] | Evaluate the intergenerational transmission of attachment in abused and neglected mothers and to examine the contributions of reflective functioning to the prediction of infant attachment disorganisation. | Longitudinal | Canada | Mother-infant dyads | 57 dyads | AAI | SSP | Trauma-specific reflective functioning measured by AAI (coded according to the Reflective-Functioning Manual for Application to AAI) | Logistic regression, chi-square test | Maternal unresolved trauma and trauma-specific reflective functioning made significant contributions to explaining the variance in infant attachment disorganisation. |
| 2. Besser & Priel, 2005 [40] | Explore the intergenerational transmission of depression and attachment insecurity across generations* | Cross-sectional | Israel | Mother-adult daughter dyads* | 100 dyads | RQ | RQ | Depression measured by the CES-D | Mediation (regression), correlation, regression | Depression in mothers led to insecure attachment orientation in daughters, with mothers' depression mediating the effect of their attachment insecurity on daughters' attachment insecurity. |
| 3. Finzi-Dottan et al., 2006 [71] | Examine the impact of parental attachment styles on the children of drug-using fathers* | Longitudinal | Israel | Drug-using father-child dyads | 56 dyads | Adult Attachment Style Questionnaire | Children's Attachment Style Questionnaire | History of drug use | Regression, ANOVA, chi-square test, correlation | Anxious/ ambivalent attachment style among drug-using fathers predicted the same attachment style in their child, while paternal avoidant attachment style correlated with secure attachment style in children. |
| 4. Iyengar et al., 2014 [41] | Examine the associations between unresolved trauma and mother and child attachment. | Longitudinal | USA | First time mothers and their infant | 47 dyads | AAI | SSP | Trauma coded in the AAI | Chi-square test, Fisher's exact test | Maternal trauma was associated with both insecure adult and infant attachment. |

(*Continued*)

**Table 4.** (Continued)

| Author/s | Aims | Design | Location | Sample | N | Adult (parent) attachment measure | Child attachment measure | Trauma/ mental health measures | Statistical analysis performed | Key results |
|---|---|---|---|---|---|---|---|---|---|---|
| 5. Karakaś et al., 2020 [69] | Investigate mother-child attachment patterns in mothers with and without a history of abuse. | Cross-sectional | Turkey | Mother-infant dyads | 94 dyads | PBI | SSP | Trauma assessed by the ACEQ | Chi-square test, Fisher's exact test | No statistically significant difference between the attachment of mothers with and without a history of abuse to their children. |
| 6. Kökçü & Kesebir, 2010 [72] | Examine the attachment pattern of parents with bipolar disorder and their children, and to investigate the relationship between attachment pattern and the clinical features of bipolar disorder* | Cross-sectional | Turkey | Mothers with bipolar disorder, their adult children, and healthy controls | 44 mothers with bipolar disorder and 35 of their children. 84 healthy controls | AAS | AAS | Bipolar disorder assessed through the SCID | Regression, correlation, ANOVA | The clinical features of bipolar disorder parents were correlated with the attachment patterns in their children. |
| 7. Lyons-Ruth et al., 2003 [73] | Examine whether maternal childhood experiences of loss or trauma contribute to maternal and child attachment. | Longitudinal | USA | High-risk mother-infant dyads | 45 dyads | AAI | SSP | Trauma coded in the AAI | Regression, correlation | The experience of parental death in a mother's childhood predicted infant attachment disorganisation at 12 months of age but not 18 months of age. |
| 8. McMahon et al., 2006 [21] | Examine the association between maternal depression and the mother-child attachment relationship. | Longitudinal | Australia | Middle class mother-infant dyads | 111 dyads | AAI | SSP | Depression measured by CED-S & Composite International Diagnostic Interview Depression Module | Regression | Mothers diagnosed with depression were more likely to have an insecure state of mind regarding attachment, and their infants were more likely to be insecurely attached, however the relationship between maternal depression and child attachment was moderated by maternal attachment state of mind. |

*(Continued)*

**Table 4.** (Continued)

| Author/s | Aims | Design | Location | Sample | N | Adult (parent) attachment measure | Child attachment measure | Trauma/ mental health measures | Statistical analysis performed | Key results |
|---|---|---|---|---|---|---|---|---|---|---|
| 9. Özcan et al., 2016 [74] | Compare attachment styles of women with psychiatric disorders and their children to a control group* | Cross-sectional | Turkey | Women with psychiatric disorders, their adult children, and healthy controls | 63 patient dyads, 63 healthy dyads | Adult Attachment Style Questionnaire | Adult Attachment Style Questionnaire | Psychiatric illness diagnosed by meeting DSM-4 criteria | Correlation, t-tests | Women with a psychiatric disorder and their children reported higher levels of insecure attachment compared to those in the control group. |
| 10. Slade et al., 2005 [46] | Examine whether maternal reflective functioning is linked to both adult and infant attachment organisation. | Longitudinal | USA | Middle class mother-infant dyads | 40 dyads | AAI | SSP | Parental reflective functioning measured by PBI | Mediation (structural equation modelling), ANOVA | Adult attachment was associated with maternal reflective functioning, and maternal reflective functioning was associated with infant attachment security. Mediation analysis revealed that adult and infant attachment was indirectly associated through reflective functioning. |
| 11. Tarabulsy et al., 2005 [20] | Examine the ecological contributions (e.g., maternal depression) to attachment transmission in a high-risk population. | Longitudinal | Canada | Adolescent mother-infant dyads | 64 dyads | AAI | AQS | Depression measured by the CED-S | Regression, ANOVA, correlation | Lower levels of maternal depression was associated with greater infant attachment security. |

*Denotes missing information not relevant to the current review (including participant groups outside of the parameters set for this review)

## Major findings

**Depression and attachment transmission.** Three articles investigated the impact of maternal depression on the intergenerational transmission of attachment. In a sample of mother-adult daughter dyads, Besser and Priel [40] found maternal depression mediated (using structural equation modelling) the relationship between mothers and children's attachment insecurity. The authors suggested that mothers' depression levels are likely associated with disturbances in the mother-child relationships resulting from less-than-optimal parenting.

Tarabulsy et al. [20] investigated the contribution of maternal depression to attachment transmission in a sample of adolescent mother-infant dyads using moderation and correlational relationship analyses. Maternal depression did not act as a moderator of the association between maternal state of mind about attachment and infant attachment security. The authors suggest this finding demonstrates the processes of attachment transmission and development are fairly stable within the context of adolescent mother-infant dyads. When examining the relationship, a direct link between maternal depression and infant attachment security was found, with lower levels of depression associated with greater infant attachment security. Thirty-five percent of the variance in infant attachment security was accounted for by maternal attachment state of mind, maternal depression, and other ecological variables (e.g., maternal education, satisfaction with paternal support).

In another study that examined the relationship between depression and attachment transmission [21], mothers diagnosed with depression were more likely to have an insecure state of mind regarding attachment, and their infants were more likely to be insecurely attached. The length of time a mother was depressed was found to impact her child's attachment security. Briefly depressed mothers were no more likely than never depressed mothers to have insecure attachment relationships with their children, while chronically depressed mothers were significantly more likely to have insecure attachment relationships with their children. The authors suggest adverse child outcomes are more likely when mothers are chronically or severely depressed. The study also tested whether maternal state of mind moderated the effect of maternal depression on child attachment. Mothers with a secure state of mind with respect to attachment, even if they experienced chronic and severe depression, were likely to have a securely attached child, while mothers with an insecure state of mind significantly increased the risk of insecure attachment, even for mothers who experienced only brief depression. The relationship between maternal depression and child attachment was, therefore, moderated by maternal attachment state of mind in this study. These findings suggest that a secure state of mind regarding attachment may buffer the effects of maternal depression on parent-child relationships.

**Trauma and attachment transmission.** Three articles investigated the impact of maternal trauma experience on the intergenerational transmission of attachment. Different types of trauma were examined across the articles, including childhood abuse and parental loss (n = 1), history of abuse or neglect (n = 1), and unresolved trauma (n = 1).

Investigating the relationship between unresolved trauma and attachment transmission, Iyengar et al. [41] compared mothers with and without unresolved trauma. Mothers with unresolved trauma were universally classified as having insecure attachment themselves compared to only 25% of mothers without unresolved trauma. Mothers with unresolved trauma were also less likely to have infants who were securely attached when compared to mothers without unresolved trauma. Notably, the study found mothers with unresolved trauma who were reorganising towards secure attachment were more likely to have a securely attached infant compared to mothers who were not reorganising. The authors conclude that the results affirm the role of unresolved trauma in the intergenerational transmission of insecure attachment and suggest that the construct of 'reorganisation' may help better predict attachment outcomes in infants of mothers with unresolved trauma.

Unlike Iyengar et al. [41], Karakas et al. [69] did not find a relationship between trauma and the intergenerational transmission of attachment. Karakas et al. [69] examined mothers with a history of abuse and neglect: almost two-thirds of mothers had experienced physical abuse, over half had experienced emotional abuse, almost a third had experienced neglect, and about 6% had experienced sexual abuse. Some mothers had experienced more than one type of abuse. Mothers with a history of abuse were less likely to have a secure attachment compared

to mothers without a history of abuse, however this difference between groups was not significant. Overall, there was no significant difference between the attachment of mothers with and without a history of abuse to their infants. The authors hypothesised that this finding may be because mothers with a history of abuse demonstrated higher level awareness during one-on-one interviews that was thought to be a result of the mothers' intention to not pass their past experiences onto the next generations.

One article found the relationship between maternal trauma experience and the intergenerational transmission of attachment differed as infants aged [73]. Lyons-Ruth et al. [73] investigated a sample of mothers who had experienced childhood abuse and/or parental loss. Almost half of mothers had experienced sexual abuse, physical abuse, or both, while 11% had witnessed serious violence between others. The study found the experience of parental death in a mother's childhood predicted infant attachment disorganisation at 12 months of age but not 18 months of age. The results, however, supported an additive model rather than a mediational model.

**Other psychiatric disorders and attachment transmission.** One article investigated attachment in women with psychiatric disorders and their children [74] while another investigated the relationship between attachment and the clinical features of bipolar disorder [72].

In the article by Özcan et al. [74], psychiatric disorder diagnoses of the women in the patient sample included major depression disorder (25.4%), bipolar disorder (41.3%), schizophrenia (28.7%), and delusional disorder (4.8%). Mothers with psychiatric disorders were more likely to have an insecure attachment compared to mothers without a psychiatric disorder. Furthermore, children of mothers with psychiatric disorders were more likely to have an insecure attachment compared to children of mothers without a psychiatric disorder, demonstrating the transmission of attachment style across generation. Notably, no significant difference was found in attachment style based on psychiatric diagnosis. Providing a potential explanation for their findings, the authors underline that having a psychiatric disorder affects a mother's care behaviors. Childcare can be neglected during the acute phases of psychiatric illness resulting in a lack of protection for the child and a deterioration in the parent-child relationship during this time, thus increasing the risk of children of mothers with psychiatric disorders developing insecure attachment.

In regard to the relationship between attachment and the clinical features of bipolar disorder, Kökcü and Keseb [72] found a reciprocal relationship between insecure attachment style and mood disorders. The study demonstrated that insecure attachment was more frequent among the parents (both mothers and fathers) with bipolar disorder and their children compared to healthy controls. Further results indicated that the risk of insecure attachment in children increased 10.2-fold and 7.5-fold if the parent with bipolar disorder was classified to have an avoidant attachment and anxious/ambivalent attachment respectively. The authors established the most significant predictor of children's insecure attachment scores was their bipolar disorder parents' own insecure attachment pattern. The findings from this study suggest that the cognitive and emotional experiences of parents with bipolar disorder may negatively affect the parent-child relationship.

**Substance use and attachment transmission.** One study examined the relationship between paternal drug use and attachment transmission [32]. The attachment classifications of drug-using fathers in the study were found to be less secure and more avoidant compared to the population norm while children of drug-using fathers were more securely attached than expected (61.8% of the sample). The authors provided a potential explanation for this finding, suggesting that the relationship between children and the non-drug using mothers may provide a barrier against the negative effect of drug-using fathers on the transmission of attachment. In the study, anxious/ambivalent attachment style among drug-using fathers predicted

the same attachment style in their child. The authors posit the difficulties in self-regulation characteristic of individuals with an anxious/ambivalent attachment style may cause drug-using fathers to seek their children's support, intensifying the anxiety of the children. Interestingly, paternal avoidant attachment style correlated with secure attachment style in children. The harmful consequences of drug-use on children may, therefore, be reduced by less involvement of drug-using fathers in their children's upbringing.

**Reflective functioning and attachment transmission.** In one study that examined reflective functioning and attachment [46], parental reflective functioning was found to play a vital role in the intergenerational transmission of attachment. In the study, adult attachment was strongly linked to maternal reflective functioning. Secure mothers had higher levels of parental reflective functioning than dismissing or preoccupied mothers. Unresolved mothers had the lowest levels of reflective function of all insecure mothers. These results suggest that mothers who were able to coherently describe their own childhood attachment experiences were more likely to be able to make sense of their children's behavior, in particular their tendencies to seek proximity, closeness, and comfort, in light of mental states. Additionally, maternal reflective functioning was linked to infant attachment security. As was the case in the analyses of adult attachment, higher levels of maternal reflective functioning were associated with secure attachment status in children, whereas lower levels of maternal reflective functioning were associated with insecure attachment status in children. Mothers of resistant and disorganised children had the lowest levels of reflective functioning. The quality of a mother's infant's attachment organisation, therefore, was strongly predicted by her capacity to reflect on her child's internal affective experience in this study. A mediational analysis (structural equation modelling) revealed that when reflective functioning was controlled for, the relationship between adult and infant attachment disappeared, suggesting that maternal reflective functioning may be a central mechanism in the intergenerational transmission of attachment. The documentation of a link between maternal reflective functioning and infant attachment in this study provides evidence to support the suggestion that a mother's capacity to make sense of her child's behaviors in light of internal, affective experience is linked to her child's feeling of safety and security in that relationship.

Another study explored the role of reflective functioning regarding trauma in the intergenerational transmission of attachment with a sample of abused and neglected mothers and their infants [45]. The authors defined reflecting functioning regarding trauma as a mother's capacity to consider traumatic experiences and their impacts in psychological terms. Unresolved trauma and reflective functioning regarding trauma made significant contributions to explaining the variance in infant attachment disorganisation. A strong concordance was found between mother and infant attachment in this study, indicative of intergenerational transmission of attachment in parents with childhood histories of abuse and neglect and their infants. Mothers with histories of abuse and high trauma-specific reflective functioning predominately had securely attached infants, while mothers with histories of abuse and low trauma-specific reflective functioning predominately had insecurely attached infants. Mothers with low trauma-specific reflective functioning were over three times more likely to have infants with insecure attachment than were mothers with histories of trauma but high trauma-specific reflective functioning. Such findings demonstrate that mothers with childhood histories of abuse and neglect struggle to respond effectively to their infant's attachment needs and foster the development of secure attachments. The authors suspect that inadequate reflective functioning of traumatic experiences may make mothers more vulnerable to transitory failures in responding consistently or regulating emotions in the context of mother–infant interactions where trauma-related emotions or memories are triggered.

## Discussion

This review aimed to systematically analyse the literature surrounding parent mental health and wellbeing and the intergenerational transmission of attachment, in order to clarify how and in what way parent mental health impacts attachment transmission. Our review found evidence that parent mental health and wellbeing play a role in the intergenerational transmission of attachment. Ten out of the eleven studies demonstrated a relationship between the parent mental health or wellbeing variable of interest and attachment transmission. This is not surprising, considering publication bias against studies with null findings in social science research [75]. The quality of the evidence, however, is mixed due to variable measurement of attachment and mental health, and lack of contemporary methodological approaches to testing associations. The review identified a number of gaps and limitations within the current literature largely pertaining to the examination of positive mental health constructs and the methods of testing mediation.

### What constructs of parent mental health are examined in the attachment transmission literature, and how are they measured?

Despite changes to the conceptualization of mental health over time, from deficit-based to wellbeing inclusive, the attachment transmission literature is largely focused on disorder and negative factors of mental health. Nine out of the eleven studies included in the review examined psychopathology constructs. Depression and trauma were the most studied mental health variables within the included studies. Psychiatric disorders and substance abuse were also examined. Interestingly, anxiety and stress, commonly investigated constructs of mental health and constructs empirically associated with attachment insecurity [76–78], did not feature in the literature. Compared to non-anxious mothers of young children, anxious mothers have been observed to be less warm and positive, more intrusive, and more critical in both clinical and community samples [79–81]. Thus, anxiety may function as a serial mediator, interfering with a parent's ability to sensitively respond to their infant resulting in insecure parent-child attachments. Parenting stress may also act as a mediating variable, with higher stress interfering with helpful socioemotional cues and affect responses that promote a secure attachment relationship developing [76].

Very few positive constructs of parent wellbeing were examined in the literature. One study each examined parental reflective functioning and trauma-specific functioning. Interestingly, although they did not measure reflective functioning directly, Karakas et al. [69] suggested this construct may be the underlying process that affected their results. No studies examined mindfulness, self-compassion, or other aspects of mentalization, as reflective functioning is a concept comparable to mentalization [82], despite strong theories these constructs could be potential transdiagnostic mediators or moderators that may explain the transmission gap [49, 53, 57, 82]. It should be noted that these constructs have been measured in relation to attachment, however not attachment transmission [83–85].

The included studies employed eight different instruments to measure seven domains of mental health. Both interview and self-report measures were used. Questionnaires and interviews were used to measure depression and trauma. Interviews were used to measure trauma-specific reflective functioning, and reflective functioning. Substance use and psychiatric disorder were measured through diagnostic assessment, while bipolar disorder was assessed through structured clinical interview. Across the studies, the measures used were reliable and valid for assessment of the respective construct.

The use of the CES-D alone, however, to measure depression in two studies may not have captured the full effect of this construct on attachment transmission. The CES-D is a measure

of depressive symptomology over the previous week [86], and therefore is unable to identify previous episode of depression or chronicity of depression. McMahon et al. [21] demonstrated that the association of maternal depression with attachment transmission differs depending on the nature of the depression (e.g., chronic, episodic, briefly depressed), using both the CES-D and structured diagnostic interview. Although Besser and Priel [40] and Tarabulsy et al. [38] found a relationship between parent depression and attachment transmission, the full effect of this relationship may not have been captured considering the use of the CES-D alone is not a good indication of a parent's experience of depression over their child's lifetime and therefore may lack diagnostic validity.

## How has attachment been measured within this literature?

Across the 11 studies, five adult attachment measures and three child attachment measures were used. This included one direct measure of each adult attachment (interview) and child attachment (observation), and six indirect (self-report) measures. Understanding how attachment has been measured within the literature is critical because the shared terminology suggests substantial overlap in meaning, however this is not supported by the research [33]. Measures of attachment are different in their emphases and correlates, and are not equally well validated [34]. For example, the concept of adult attachment assessed in self-report measures typically relates to the expectations and perceptions individuals hold about themselves and their close relationships [87]. Evidently assessing something different, the AAI, a direct measure of attachment, accesses internal working models of childhood caregiving experiences [14]. Perhaps the fundamental difference between self-report measures and the AAI is that self-report measures are affected by bias while the AAI is designed to 'surprise the unconscious' [14]. Thus, differences in measurement likely introduce variability in the latent construct that is being assessed.

Despite the range of attachment measures used within the literature, and therefore the latent construct being assessed, findings did not appear to differ based on the measures employed. Studies that utilised direct and/or indirect measures all found evidence supporting the role of parent mental health in the intergenerational transmission of attachment, with one exception. Using the PBI to assess adult attachment, Karakas et al. [69] found no significant difference between the attachment of mothers with and without a history of abuse to their children. The PBI is a measure of parenting behaviors however it was used in this study to evaluate a mother's attachment with their own parent, likely accounting for the absence of a relationship demonstrated by other research outlined in this review.

To build strong evidence supporting the role of mental health and wellbeing and other mechanisms in the intergenerational transmission of attachment, future research must rely on the use of valid, reliable and sensitive attachment measures. Thus, the AAI and SSP should be relied on as the primary measures of attachment considering these measures: are direct measures of attachment; are the most established measures of attachment; have excellent psychometric properties; and provide the most detailed data to assess attachment transmission [33]. Use of the AAI and SSP brings its own challenges. Both measures require significant resources, time, training for administration, and coding [33]. Furthermore, the SSP is validated for use with infants aged 9 to 20 months [5] while the AAI is for use with adults [14]. Measurement of attachment in childhood and adolescence is a field of continued development. The attachment field is dependent on the availability of valid, reliable, and sensitive measures in order to test theories and build evidence. The further development of psychometrically sound measures of attachment in middle childhood and adolescence will help to advance the attachment literature and understand the transmission gap.

## What statistical models have been used within this literature?

Statistical models used within the literature to determine how parent mental health and wellbeing impacts the intergenerational transmission of attachment were: mediation; moderation by parent mental health; moderation by parent attachment; and correlations within a specific high-risk sample. The majority of the literature used correlations within a specific high-risk population rather than directly testing mediation or moderation to investigate the relationship between parent mental health and attachment transmission. While identifying that a relationship exists between parent mental health and the transmission of attachment is valuable, correlation does not identify and explain the mechanism or process that underlies the relationship between these two variables, which is necessary to close the transmission gap. Helping to clarify the nature of the relationship between parent mental health and attachment transmission, two studies used mediation analyses and two studies used moderation analyses (one moderation by parent mental health and one moderation by parent attachment).

One out of the two studies that tested mediation did so using the causal steps regression method proposed by Baron and Kenny [59]. While previously the most widely used method to test mediation, significant issues with this approach to mediation testing have become clear in recent years [88]. Issues of concern include reduced statistical power [89, 90], increased likelihood of both Type I and Type II errors [89], and the lack of a formal test of the significance of the mediated pathway [91]. We and other authors [92] encourage researchers to move beyond Baron and Kenny's approach and to utilise more robust statistical methodology such as structural equation modelling or bootstrapping approaches.

Bootstrapping approaches are advantageous because that are more robust to smaller sample sizes [61], an issue in attachment research due to the laborious methodology. As highlighted by Verhage et al. [28], in order to show the intergenerational transmission of attachment affects, substantial samples are required. Verhage et al. [28] report samples of over 180 caregiver-child dyads are required when researchers want to look at full range of attachment classifications instead of an isolated examination of secure versus insecure contrasts, or when studying at risk samples. The current review investigated at-risk samples, with the largest sample being 111 parent-child dyads. Thus, it is likely the studies included in this review were underpowered which reduces the chance of detecting a true effect and also reduces the likelihood that a statistically significant result reflects a true effect [93].

## Can the relationship between adult attachment and child attachment be explained by mediation and/or moderation models using parent mental health?

This review has established that there is evidence to suggest that the relationship between adult attachment and child attachment can be better understood via the inclusion of parent mental health and wellbeing as a mediator variable. The two studies that investigated the mediation model each demonstrated that parent mental health explains at least some of the relationship between adult attachment and child attachment. In particular, parental depression and reflective functioning were each found to be mechanisms through which adult attachment was associated with child attachment. Thus, it can be concluded that some of the unexplained variance in the transmission of attachment can be explained by parent mental health and wellbeing. There is not enough evidence in the literature to draw conclusions about whether the relationship between adult attachment and child attachment can be explained by moderation using parent mental health, as only one study utilised this model. While this review has identified parental mental health and wellbeing as a credible mechanism of attachment transmission, it

also recognises that further research and evidence is necessary to draw conclusions with any certainty.

## Implications for future research

The findings of this review highlight important considerations for future research. Firstly, it is obvious that future research investigating attachment transmission must extend beyond examining disorder and deficits in mental health and more widely examine constructs related to positive wellbeing. Particularly, mindfulness, self-compassion, and mentalization are promising potential mechanisms in the intergenerational transmission of attachment. Secondly, future attachment transmission research would benefit by being conducted in a more structured way. There is sufficient evidence to conclude that a relationship exists between parent mental health and wellbeing and the transmission of attachment. Research must now move towards investigating the precise mechanisms of how and why parent mental health and wellbeing affects the intergenerational transmission of attachment. Particularly, research must conduct mediation and moderation analyses using contemporary statistical techniques. Finally, future research could benefit from utilising archival data. A number of the articles screened in the full-text review component of this review included all of the variables necessary to conduct mediation and moderation analyses and investigate the intergenerational transmission of attachment but yet did not [94–96]. Thus, archival research is a plausible, not to mention less expensive and labour-intensive option, to extend the literature.

## Clinical implications

Although all types of insecure attachment occur in community samples, clinical populations are at increased risk, especially for intergenerational transmission of disorganised attachment [15]. Understanding the variables that affect the intergenerational transmission of attachment can inform the development of interventions that can mitigate attachment issues. For example, as evidenced by this review, parental reflective functioning impacts attachment transmission, with higher levels of reflective functioning associated with secure attachment in both parents and infants. With this finding in mind, clinical interventions may benefit by using approaches that increase a parent's reflective functioning abilities in order to improve relational and parenting outcomes. The further examination of the impact of positive wellbeing constructs on attachment transmission is particularly important clinically. Parent mindfulness and self-compassion are also modifiable through intervention [51, 97–99]. If we can understand how constructs such as mindfulness and self-compassion are related to attachment transmission, clinical practice approaches can be tailored from the findings which would result in positive parent-child relationship outcomes.

## Strengths/Limitations

To our knowledge, this is the first systematic review to address the role of parent mental health in understanding attachment transmission. The review was registered in PROSPERO before it was commenced, and it was carried out in accordance with PRISMA guidelines (http://prisma-statement.org/). Comprehensive searches were conducted in large databases.

A number of important limitations need to be addressed. A weakness in any systematic review is that the interpretation of the findings is dependent on the quality and scope of the included studies. A specific limitation for this review is the small body of research examining how impact of parent mental health and wellbeing might impact the intergenerational transmission of attachment as well as the small sample sizes of the included studies, which makes it difficult to draw strong conclusions. Additionally, the use of Baron and Kenny's [59] approach

to mediation testing in one out of the two studies that investigated this model raises concerns about the effects of random measurement errors. The mediating role of parent mental health in the relation between adult and child attachment found may have been under- or over-estimated in the study by Besser and Priel [40] because of the statistical technique used (regression method), which is subject to the increased likelihood of both Type I and Type II errors [100]. Finally, although we completed a systematic search of the relevant literature, it is possible that we screened out or failed to include potentially relevant publications.

## Conclusion

This systematic review found that parent mental health and wellbeing impacts the intergenerational transmission of attachment. However, it has also highlighted that there are currently large gaps within the current literature that need to be addressed in order for the relationship between parent mental health and wellbeing and attachment transmission to be better understood. The limited scope of parent mental health and wellbeing constructs examined in the literature, the sparse use of robust statistical analyses (e.g., mediation, moderation), and the lack of literature in general makes it difficult to draw conclusions on how and why parent mental health and wellbeing impacts attachment transmission. These limitations must be addressed and rectified in future research so that progress can be made in understanding the implications of parental mental health and wellbeing on the intergenerational transmission of attachment.

## Supporting information

**S1 Checklist. PRISMA checklist.**
(DOC)

**S1 File. PROPSERO protocol (registration number: CRD42020157247).**
(PDF)

## Author Contributions

**Conceptualization:** Alixandra Risi, Judy A. Pickard, Amy L. Bird.

**Data curation:** Alixandra Risi.

**Formal analysis:** Alixandra Risi, Judy A. Pickard, Amy L. Bird.

**Investigation:** Alixandra Risi, Amy L. Bird.

**Methodology:** Alixandra Risi, Judy A. Pickard, Amy L. Bird.

**Project administration:** Alixandra Risi.

**Supervision:** Judy A. Pickard, Amy L. Bird.

**Writing – original draft:** Alixandra Risi.

**Writing – review & editing:** Alixandra Risi, Judy A. Pickard, Amy L. Bird.

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
