## [Decision Letter · Decision Letter 0]

2 Sep 2021

PONE-D-21-20268

The impact of parent mental health on the intergenerational transmission of attachment: A systematic review

PLOS ONE

Dear Dr. Risi,

Thank you for submitting your manuscript to PLOS ONE. After careful consideration, we feel that it has merit but does not fully meet PLOS ONE’s publication criteria as it currently stands. Therefore, we invite you to submit a revised version of the manuscript that addresses the points raised during the review process.

I was fortunate to receive reviews from two reviewers with expertise on attachment. The two reviews differed a great deal regarding the potential impact of the manuscript, with the first reviewer suggesting rejection and the second reviewer suggesting minor revisions. 

If you choose to submit a revised manuscript, please address the significant issues noted by reviewer 1 by providing a strong rationale for examining psychopathology as a mediator and addressing differences in attachment measures.

We look forward to receiving your revised manuscript.

Kind regards,

Soo Hyun Rhee, Ph.D.

Academic Editor

PLOS ONE

Journal Requirements:

2. We note that you have referenced (ie. Bewick et al. [5]) which has currently not yet been accepted for publication. Please remove this from your References and amend this to state in the body of your manuscript: (ie “Bewick et al. [Unpublished]”) as detailed online in our guide for authors

3. We note that this manuscript is a systematic review or meta-analysis; our author guidelines therefore require that you use PRISMA guidance to help improve reporting quality of this type of study. Please upload copies of the completed PRISMA checklist as Supporting Information with a file name “PRISMA checklist

4. PLOS ONE does not copy edit accepted manuscripts (https://journals.plos.org/plosone/s/criteria-for-publication#loc-5). To that effect, please ensure that your submission is free of typos and grammatical errors.

*Please note that your quality assessment table (Table 1) appears to be incomplete. During your revision, please include a completed version.

Reviewers' comments:

Reviewer's Responses to Questions

**Comments to the Author**

1. Is the manuscript technically sound, and do the data support the conclusions?

Reviewer #1: No

Reviewer #2: Yes

2. Has the statistical analysis been performed appropriately and rigorously? 

Reviewer #1: Yes

Reviewer #2: Yes

3. Have the authors made all data underlying the findings in their manuscript fully available?

Reviewer #1: Yes

Reviewer #2: Yes

4. Is the manuscript presented in an intelligible fashion and written in standard English?

Reviewer #1: Yes

Reviewer #2: Yes

5. Review Comments to the Author

Reviewer #1: This paper proposes to examine the roles of psychopathology as a factor in explaining cross-generational transmission of attachment security/insecurity. The issue is a significant one and there is as yet no explanation for the low level of cross-generational transmission.

Unfortunately the paper has three significant problems:

1. It is not obvious that psychopathology is the place to look for additional predictive power across generations here. If it turned out that psychopathology contributed to predicting attachment across generations it would not really be a specific contribution to attachment theory. It would just be a generic effect that the disorganization associated with psychopathology has on so many other variables. For example, psychopathology might account in part for inconsistency in arithmetic scores from year to year. What attachment theorists are looking for would be something specifically related to attachment processes, e.g., contribution from the other parent, marital variables, additional demands on the mother such as employment, etc. Something that goes to the processes the theory implicates in attachment development. Even if psychopathology were statistically a significant factor (see below) it would simply raise the question what aspects of caregiving mediate the effect. At that point we would be right back where we are because the problem in not being able to predict attachment security across-generations seems to lie in not knowing what aspects of care (other than sensitivity) are causal. In brief, even if the study had found significant effects related to psychopathology, this would be of limited interest to attachment theorists.

2. The authors find only a few studies from which to extract data. This probably reflects the point mentioned above. Moreover, the few studies use a variety of different attachment measures. The adult measures, in particular, as absolutely not comparable. Despite being entitles "attachment measures" by the people who developed them, there is no evidence that they measure the same constructs. Indeed, there is considerable evidence that they are barely if at all inter-correlated and even this small association is easily accounted for by nuisance variables. In brief, having searched for studies on which to conduct their study, the authors could have determined without any statistical analysis at all that the literature does not support any useful conclusions regarding psychopathology as a mediator of cross-generation attachment transmission.

3. Attachment theorists have gone to some lengths to figure out why cross-generation transmission of attachment security is not as strong as they expect. Numerous studies have examined the issue. In addition, there are numerous conceptual analyses of the problem. Importantly, almost all of this literature refers to work on populations in which psychopathology occurs at very low levels. While one might find some additional significant cross-generational variance in samples that are more diverse with respect to psychopathology, as mentioned above, this would simply reflect generic (rather than attachment specific) effects. It would have little relevance to the problem attachment researchers have set themselves, which is to understand why there is not more cross-generation transmission in ordinary samples.

In conclusion, the study is not well designed to address the issue of cross-generational transmission of attachment security. It is hard to agree with the authors that new research (which would entail samples with diversity in psychopathology) deserves high priority. More useful would be new (empirically buttressed) insights into the factors that contribute to attachment development. Even then, the issue will be difficult to resolve because it will always be difficult to distinguish between low consistency arising from design and measurement weaknesses and low consistency that would arise if the answer to the question were that attachment theorists are wrong in predicting greater consistency across generations.

Sorry I can't be more positive about the study. It is a difficult issue to address and attachment researchers have not provided what the authors needed.

Reviewer #2: PONE-D-21-20268

The impact of parent mental health on the intergenerational transmission of attachment: A systematic review

The present study aims i) to review the association between parent’s mental health and the intergenerational transmission of attachment provided in 11 published studies, and ii) to add recommendations for research, and clinical practice. The study addresses an important gap in attachment intergenerational transmission. The manuscript is very well written, the framework is very appropriated, the statistical analyses are performed to a high technical standard, and the results are interesting and relevant although there are important limitations addressed by the authors regarding the studies sample size, the robustness of analyses or diversity of methodological options.

Although the manuscript is quite long, I do not recommend an “abbreviated” version. In act, find all information relevant and enlightening.

I have two minor suggestions:

1. In table 4 (p.21-24), I believe it is relevant to include information about the statistical analysis performed in each study. Particularly, because the authors discuss this topic.

2. In the abstract, the authors claim that is difficult to draw conclusions based on the published articles, I agree. However, I think they should alight the most significant results and directions found in their review, otherwise, the readers lose interest after reading the abstract. For instance, "Thus, anxiety may function as a serial mediator, interfering with a parent’s ability to sensitively respond to their infant resulting in insecure parent-child attachments." (p. 38). As well as results on parental substance abuse, depression or trauma.

Overall, the study gives an important contribution to future research on the attachment field and explores the associations between parental mental health and attachment transmission. The systematic review is well performed and enlightening.

I wish the authors good luck in their future research.

6. PLOS authors have the option to publish the peer review history of their article (what does this mean?). If published, this will include your full peer review and any attached files.

Reviewer #1: No

Reviewer #2: No

---

## [Author Response · Author response to Decision Letter 0]

24 Oct 2021

Reviewer 1

Point 1: It is not obvious that psychopathology is the place to look for additional predictive power across generations here. If it turned out that psychopathology contributed to predicting attachment across generations it would not really be a specific contribution to attachment theory. It would just be a generic effect that the disorganization associated with psychopathology has on so many other variables. For example, psychopathology might account in part for inconsistency in arithmetic scores from year to year. What attachment theorists are looking for would be something specifically related to attachment processes, e.g., contribution from the other parent, marital variables, additional demands on the mother such as employment, etc. Something that goes to the processes the theory implicates in attachment development. Even if psychopathology were statistically a significant factor (see below) it would simply raise the question what aspects of caregiving mediate the effect. At that point we would be right back where we are because the problem in not being able to predict attachment security across-generations seems to lie in not knowing what aspects of care (other than sensitivity) are causal. In brief, even if the study had found significant effects related to psychopathology, this would be of limited interest to attachment theorists.

The first point raised by Reviewer 1 questions whether examining psychopathology as a mediator for attachment transmission is useful and suggests that if psychopathology did contribute to predicting attachment across generations, it would not contribute significantly to attachment theory. We apologise for our lack of clarity which lead to confusion about the scope of this study. To clarify, through our systematic review, we were investigating the current literature surrounding the implications of parent mental health and wellbeing on parent-child attachment. We can understand how especially Figure 1 and the explanations related to that could read as an attempt to reconceptualise attachment theory. We still believe that it is important to understand the role of mental health and wellbeing in attachment transmission. While much observational research has been conducted with community samples, we must also understand how attachment transmission presents in non-community samples as this is where much of attachment-informed intervention work occurs. In addition, the interventions for the treatment of parental mental health as opposed to dyadic interaction are quite distinct. Therefore, if psychopathology and positive mental health constructs were found to play any role in attachment transmission, this could have significant clinical implications. We recognise some of our wording used throughout the manuscript was unclear about the scope of our study. We have now modified the title of our manuscript to de-emphasise the theoretical component of the study. The revised title reads ‘The implications of parent mental health and wellbeing on parent-child attachment: A systematic review’. To provide increased clarity about the role mental health and wellbeing may play in the relationship between parent and infant attachment, we have removed Figure 1 and updated Figure 2 (now Figure 1). This figure better proposes that the potential mediating or moderating role of mental health or wellbeing is transmitted through differences in caregiving behavior. Finally, we have reduced the emphasis of our systematic review ‘closing the transmission gap’ by modifying our language throughout the manuscript (e.g., see p.2, p.11, pp.38-39).

Additionally, Reviewer 1 suggests that if psychopathology did contribute to predicting attachment transmission, it would be due to the relationship between psychopathology and attachment disorganisation. We agree that psychopathology and attachment disorganisation are related. However, we also think that viewing psychopathology and attachment disorganization as distinct constructs may have some value in the following ways:

• Psychopathology and disorganisation are related differentially to child outcomes. The distinction between psychopathology and disorganisation is demonstrated by longitudinal research showing psychopathology is a correlate of continuity and discontinuity in attachment. Results stemming from the Minnesota Longitudinal Study of Parents and Children demonstrate disruptions in attachment across time can be associated with parental psychopathology (e.g., Sroufe, 2005; Sroufe, Coffino, & Carlson, 2010; Sroufe, Eyeland, & Kreutzer, 1990; Weinfield, Sroufe, & Egeland, 2000). Changes in parental stress and/or depression were found to relate to continuity and change in attachment classification from infancy to adulthood in a high-risk sample. Individuals with secure early attachment histories may show a rebound to better functioning after setbacks (e.g., parental stress/depression decreases). These findings highlight the distinction between psychopathology and attachment classification and provide a strong rationale for examining parental psychopathology as a mediator in attachment transmission.

• It is possible to have episodes of psychopathology without necessarily leading to attachment disorganisation, and vice versa. For example, an individual may experience episodes of depression or may have a genetic predisposition to schizophrenia, however these do not automatically, in themselves, infer attachment disorganisation. For example, Riggs et al. (2007) found both proportions of secure and insecure AAI classifications were represented in a psychiatric sample of trauma survivors. 

We would also like to highlight that our systematic review does not focus solely on examining disorganised versus organised attachment and considers all attachment classifications. It also extends beyond psychopathology and considers mental wellbeing to examine the full continuum of mental health. Examining parent wellbeing constructs allows us to consider how the internal resources of the parent (e.g., mindfulness, reflective functioning) relate to attachment processes. Reviewer 1 comments that even if psychopathology was a statistically significant factor, it would simply raise the question what aspects of caregiving mediate the effect. As demonstrated in the results and discussion of our systematic review, reflective functioning and other positive mental health constructs may mediate this effect. Mental health lies along a continuum with positive wellbeing and the consideration of these variables may provide answers to some of the queries raised by Reviewer 1. Upon reflection, we acknowledge we did not emphasize the inclusion of mental wellbeing enough in our initial manuscript. We have modified our manuscript to ensure parent mental health and wellbeing are equally emphasised throughout. For example, instead of using only ‘parent mental health’ when talking about the constructs examined throughout our systematic review, we now use the phrase ‘parent mental health and wellbeing’. To ensure it is clear to readers that we are examining the complete continuum of mental health in our systematic review, we have also modified the title of our manuscript to include ‘wellbeing’. 

Point 2: The authors find only a few studies from which to extract data. This probably reflects the point mentioned above. Moreover, the few studies use a variety of different attachment measures. The adult measures, in particular, as absolutely not comparable. Despite being entitles "attachment measures" by the people who developed them, there is no evidence that they measure the same constructs. Indeed, there is considerable evidence that they are barely if at all inter-correlated and even this small association is easily accounted for by nuisance variables. In brief, having searched for studies on which to conduct their study, the authors could have determined without any statistical analysis at all that the literature does not support any useful conclusions regarding psychopathology as a mediator of cross-generation attachment transmission.

We are in full agreement with Reviewer 1 that the variety of attachment measures used across the studies included in the systematic review do not measure the same constructs. We recognise that although a variety of measures may be described as attachment measures, evidence suggests they are not actually measuring the same construct. The literature, however, has not always clearly made this distinction. As this is a systematic review and not a meta-analysis, it is important that all studies which report they are measuring ‘attachment’ be included in the review. In the beginning stages of our systematic review, we did consider only examining literature that used gold-standard measures of attachment. However, we thought it would be helpful for readers to understand that this body of literature utilises gold-standard measures as well as indirect measures that may reflect a different construct. The inclusion of all studies in our systematic review allowed us draw conclusions about the state of the literature being limited due to the variety of measures which capture different constructs. To provide clarity for future readers, we have made several revisions in our discussion on the differences in attachment measures. Specifically, we have extended our discussion regarding measurement differences in the Introduction (see pp.5-6), provided further justification for including all measures in the Method (see p.13), and elaborated on the importance of understanding how attachment has been measured in the literature in the Discussion (see p.33).

Point 3: Attachment theorists have gone to some lengths to figure out why cross-generation transmission of attachment security is not as strong as they expect. Numerous studies have examined the issue. In addition, there are numerous conceptual analyses of the problem. Importantly, almost all of this literature refers to work on populations in which psychopathology occurs at very low levels. While one might find some additional significant cross-generational variance in samples that are more diverse with respect to psychopathology, as mentioned above, this would simply reflect generic (rather than attachment specific) effects. It would have little relevance to the problem attachment researchers have set themselves, which is to understand why there is not more cross-generation transmission in ordinary samples.

In their final point, Reviewer 1 states attachment researchers are interested in understanding why there is not more cross-generational transmission of attachment in community/low-risk samples, and thus, suggests our systematic review will not make a significant contribution considering psychopathology occurs in low levels in such samples. We recognise that the investigation of attachment transmission across all samples is important. However, we believe the examination of attachment transmission in high-risk samples is also of importance. Indeed, several important attachment findings have come from high-risk samples (e.g., Sroufe, 2005; Weinfield, Sroufe, & Egeland, 2000). As noted above, attachment-based interventions are also focused on high-risk and clinical samples because higher rates of insecure attachment occur in these populations. Understanding the variables that affect attachment can inform the development of interventions that can mitigate attachment issues, emphasizing the importance of our review.

Reviewer 2

Point 1: In table 4 (p.21-24), I believe it is relevant to include information about the statistical analysis performed in each study. Particularly, because the authors discuss this topic.

We thank reviewer two for this suggestion and have now included information about the statistical analyses performed in each study in Table 4. 

Point 2: In the abstract, the authors claim that is difficult to draw conclusions based on the published articles, I agree. However, I think they should alight the most significant results and directions found in their review, otherwise, the readers lose interest after reading the abstract. For instance, "Thus, anxiety may function as a serial mediator, interfering with a parent’s ability to sensitively respond to their infant resulting in insecure parent-child attachments." (p. 38). As well as results on parental substance abuse, depression or trauma.

We have made modifications to the results section of the Abstract with Reviewer 2’s suggestions in mind to highlight the most significant results and directions found in the review.

---

## [Decision Letter · Decision Letter 1]

19 Nov 2021

The implications of parent mental health and wellbeing for parent-child attachment: A systematic review

PONE-D-21-20268R1

Dear Dr. Risi,

We’re pleased to inform you that your manuscript has been judged scientifically suitable for publication and will be formally accepted for publication once it meets all outstanding technical requirements.

Kind regards,

Soo Hyun Rhee, Ph.D.

Academic Editor

PLOS ONE

Additional Editor Comments (optional):

Reviewers' comments:

Reviewer's Responses to Questions

**Comments to the Author**

1. If the authors have adequately addressed your comments raised in a previous round of review and you feel that this manuscript is now acceptable for publication, you may indicate that here to bypass the “Comments to the Author” section, enter your conflict of interest statement in the “Confidential to Editor” section, and submit your "Accept" recommendation.

Reviewer #2: All comments have been addressed

Reviewer #3: All comments have been addressed

2. Is the manuscript technically sound, and do the data support the conclusions?

Reviewer #2: Yes

Reviewer #3: Yes

3. Has the statistical analysis been performed appropriately and rigorously? 

Reviewer #2: Yes

Reviewer #3: N/A

4. Have the authors made all data underlying the findings in their manuscript fully available?

Reviewer #2: Yes

Reviewer #3: Yes

5. Is the manuscript presented in an intelligible fashion and written in standard English?

Reviewer #2: Yes

Reviewer #3: Yes

6. Review Comments to the Author

Reviewer #2: After reading the new version of the manuscript, I find that the authors fully addressed all my concerns.

Reviewer #3: The authors had already undergone a revision round with two external reviewers. I think they addressed thoroughly each and every comment and I found this review useful and sound. I agree with the concerns previously expressed by the Reviewers and the authors have properly responded to them.

I have no other concern.

7. PLOS authors have the option to publish the peer review history of their article (what does this mean?). If published, this will include your full peer review and any attached files.

Reviewer #2: No

Reviewer #3: No

---

## [Editor Report · Acceptance letter]

25 Nov 2021

PONE-D-21-20268R1 

The implications of parent mental health and wellbeing for parent-child attachment: A systematic review 

Dear Dr. Risi:

I'm pleased to inform you that your manuscript has been deemed suitable for publication in PLOS ONE. Congratulations! Your manuscript is now with our production department. 

Kind regards, 

on behalf of

Dr. Soo Hyun Rhee 

Academic Editor

PLOS ONE